# Transcription dynamically patterns the meiotic chromosome-axis interface

Xiaoji Sun[1†‡], Lingzhi Huang[2†], Tovah E Markowitz[1], Hannah G Blitzblau[3], Doris Chen[2], Franz Klein[2*], Andreas Hochwagen[1*]

[1]Department of Biology, New York University, New York, United States; [2]Max F. Perutz Laboratories, University of Vienna, Wien, Austria; [3]Whitehead Institute for Biomedical Research, Cambridge, United States

**Abstract** Meiotic chromosomes are highly compacted yet remain transcriptionally active. To understand how chromosome folding accommodates transcription, we investigated the assembly of the axial element, the proteinaceous structure that compacts meiotic chromosomes and promotes recombination and fertility. We found that the axial element proteins of budding yeast are flexibly anchored to chromatin by the ring-like cohesin complex. The ubiquitous presence of cohesin at sites of convergent transcription provides well-dispersed points for axis attachment and thus chromosome compaction. Axis protein enrichment at these sites directly correlates with the propensity for recombination initiation nearby. A separate modulating mechanism that requires the conserved axial-element component Hop1 biases axis protein binding towards small chromosomes. Importantly, axis anchoring by cohesin is adjustable and readily displaced in the direction of transcription by the transcriptional machinery. We propose that such robust but flexible tethering allows the axial element to promote recombination while easily adapting to changes in chromosome activity.

**\*For correspondence:** andi@nyu.edu (AH); franz.klein@univie.ac.at (FK)

†These authors contributed equally to this work

**Present address:** ‡Sackler Institute of Graduate Biomedical Sciences, School of Medicine, New York University, New York, United States

**Competing interests:** The authors declare that no competing interests exist.

## Introduction

Meiosis is a specialized developmental program in which a diploid cell undergoes two nuclear divisions to produce the haploid gametes required for sexual reproduction. A key event in meiosis is the programmed recombination of homologous chromosomes, which ensures proper chromosome segregation during the first meiotic division and also provides genetic variation in the offspring. Homologous recombination events initiate with programmed DNA double-strand breaks (DSBs) that are introduced by the topoisomerase-like enzyme Spo11. Upon endonucleolytic removal of Spo11 from break ends, a subset of DSBs are repaired by crossing-over thereby physically linking homologous chromosome pairs for segregation during meiosis I. The genomic distribution of meiotic DSBs is decidedly non-random (*Gerton et al., 2000*; *Blitzblau et al., 2007*; *Buhler et al., 2007*; *Pan et al., 2011a*), but the mechanisms driving this distribution, and thus the patterns of recombination, are not yet well defined.

DSB formation and repair occur in the context of a specific chromosome architecture characterized by linear arrays of chromatin loops with the bases attached to a proteinaceous axis, known as the axial element (*Moens and Pearlman, 1988*; *Zickler and Kleckner, 1999*). In *Saccharomyces cerevisiae* most DSBs occur within the chromatin loops, whereas axis association sites are cold for DSB formation (*Blat et al., 2002*; *Panizza et al., 2011*; *Ito et al., 2014*). The axial element in yeast contains the axis proteins Red1 and Hop1 (*Hollingsworth et al., 1990*; *Smith and Roeder, 1997*), as well as the meiotic cohesin complex (*Klein et al., 1999*). Red1 and Hop1 physically interact with each other (*de los Santos and Hollingsworth, 1999*), and Red1 helps recruit Hop1 to chromosomes (*Smith and Roeder, 1997*). Meiotic cohesin is not essential for chromosomal binding of Red1 and Hop1, but Red1 and Hop1 distribution is

**eLife digest** Chromosomes are long molecules of DNA that represent the genetic material of an organism. In most animal cells, chromosomes are found in pairs; with one inherited from the mother and the other from the father. Sex cells—egg cells and sperm—contain half the normal number of chromosomes, so that when they fuse, the resulting single-celled embryo inherits the full set.

When sex cells are being produced, a ring made from a protein called cohesin encircles each pair of chromosomes and holds them together until they are ready to be separated. The paired chromosomes also swap sections of DNA via a process called recombination. Structures, referred to as axial elements, compact the chromosomes in each pair and bring them in close contact so that recombination can take place. In the sexually reproducing baker's yeast, axial elements contain three main proteins: cohesin, Hop1, and Red1, but it remains unclear how the entire structure is anchored to the underlying chromosomes. Furthermore, the genes encoded within the DNA of the compacted chromosomes remain active, but it is also not clear how this is possible. This is because the compacted structure would be expected to prevent the molecular machinery that expresses genes from accessing the DNA.

Sun, Huang et al. have now studied this process in budding yeast cells by using a method called ChIP-seq to determine where cohesin and the Hop1 and Red1 proteins are found along the chromosomes. The experiments showed that cohesin, Hop1, and Red1 are enriched in regions between two genes that run in the opposite directions to each other. Sun, Huang et al. also observed that cohesin recruits Red1, which in turn, recruits Hop1, and that all three proteins physically interact with one another. These findings imply that it is cohesin that anchors the axial elements to the underlying chromosomes.

Further experiments showed that cohesin slides along chromosomes towards areas where genes are active. This suggests that cohesin provides a robust, but flexible, link between the axial elements and the chromosomes. This flexibility would enable recombination and gene expression to continue in compacted chromosomes. A loss of flexibility may be one of the reasons why mutations in cohesin components of the axial element cause infertility in men and condition called premature ovarian failure in women.

highly abnormal in cohesin mutants (*Klein et al., 1999*; *Panizza et al., 2011*). Notably, this phenotype is not due to a loss of sister chromatid cohesion, as axis protein distribution is largely normal on unreplicated chromosomes (*Blitzblau et al., 2012*). It is currently unclear whether cohesin directly interacts with Red1 and/or Hop1 in budding yeast.

One of the first functions of Red1 and Hop1 is to correctly localize the essential DSB factors Mer2, Rec114 and Mei4 to axis sites (*Panizza et al., 2011*). Accordingly, DSB levels are strongly reduced in *hop1Δ* or *red1Δ* mutants (*Mao-Draayer et al., 1996*; *Schwacha and Kleckner, 1997*; *Xu et al., 1997*). Moreover, reduction in Red1 and Hop1 binding in the absence of cohesin correlates with reduced Spo11 binding and low DSB levels on several larger chromosomes (*Kugou et al., 2009*; *Panizza et al., 2011*). Conversely, increased binding of axis proteins on small chromosomes correlates with increased DSB levels and crossover recombination (*Kaback et al., 1992*; *Blitzblau et al., 2007*; *Pan et al., 2011a*; *Panizza et al., 2011*).

The molecular determinants of axis attachment sites remain poorly defined. Red1 was reported to favor AT-rich segments on chromosome III at a resolution of 3 kb (*Blat et al., 2002*), whereas Hop1 has preferential affinity for G-rich sequences and specific DNA topologies in vitro (*Kironmai et al., 1998*). However, on meiotic chromosomes, Hop1 and Red1 binding coincides with cohesin binding sites and regular distribution of Hop1 requires cohesin (*Panizza et al., 2011*), suggesting that cohesin is a major determinant of axis attachment sites. The yeast meiotic cohesin complex consists of Smc1, Smc3, Scc3, and the kleisin Rec8, which replaces the mitotic kleisin Scc1/Mcd1 (*Klein et al., 1999*; *Watanabe and Nurse, 1999*). Despite the different subunit composition, the cohesin-associated chromosomal sites are highly conserved between mitosis and meiosis (*Blat and Kleckner, 1999*; *Glynn et al., 2004*). A strong correlation between cohesin sites and regions of convergent transcription has been observed (*Lengronne et al., 2004*), and changes in transcriptional activities can result in altered cohesin localization (*Lengronne et al., 2004*; *Bausch et al., 2007*). Because cohesin

can encircle DNA from two sister-chromatids (*Ivanov and Nasmyth, 2005*, *2007*) it is hypothesized to be pushed along genes ahead of the RNA polymerase II (RNAPII) complex, thus accumulating at convergent regions (*Ocampo-Hafalla and Uhlmann, 2011*). However, sliding has not been addressed in meiotic prophase, where cohesin may be stably incorporated in the axial element. Moreover, the mechanisms of meiotic axis protein loading and the functional link to cohesin remain to be elucidated.

Here, we probed axis assembly using high-resolution chromatin binding studies of axis protein distribution and in vivo detection of physical associations between axis proteins. We found that axis proteins dynamically accumulate between convergent genes in a distinctive two-peak pattern, which we demonstrate to be a direct consequence of the underlying transcriptional activity. We show that cohesin is responsible for this behavior and that Rec8 interacts closely with both axis proteins via Red1. In the absence of Rec8, Hop1 becomes essential for chromosomal Red1 recruitment, defining a cohesin-independent mode of axis protein recruitment. Intriguingly, Hop1 but not Rec8 is required for the preferential loading of Red1 to short chromosomes, possibly by preventing excess Red1 accumulation on large chromosomes and at centromeres. Our findings demonstrate that the well-dispersed binding and chromosome size-biased enrichment of axis proteins are controlled by two independent recruitment modes and provide important insight into the mechanistic hierarchy of meiotic axis assembly.

## Results

### Genome-wide distribution of axis association sites

We used chromatin immunoprecipitation followed by deep sequencing (ChIP-seq) to determine the chromosomal distribution of two axis proteins, Red1 and Hop1, as well as two cohesin subunits, Rec8 and Smc3. To validate our data, we compared the resulting profiles with previous lower-resolution ChIP–chip data of Hop1, Red1, and Rec8 (*Blitzblau et al., 2012*). A high correlation indicated that the identification of axis association sites was consistent between the two approaches (Pearson's $r = 0.80$). As observed previously, the genome-wide distribution of axis proteins and cohesin was highly correlated (*Figure 1—figure supplement 1A,B*), with the exception of centromeric regions where cohesin was more highly enriched than axis proteins. Because of the high correspondence in localization, we chose the Red1 data set as the representative axis protein data set for subsequent analyses.

We observed Red1 binding sites distributed across all chromosomes, with a notable bias toward the smallest chromosomes. Statistical analysis (*Zhang et al., 2008*) derived a total of 774 Red1 binding sites, in close agreement with previous reports, which identified 656 and 802 axis association sites, respectively (*Panizza et al., 2011*; *Blitzblau et al., 2012*). Binding was particularly strong on the three shortest chromosomes, which displayed a higher overall level of axis protein enrichment (*Figure 1—figure supplement 1C*) (*Panizza et al., 2011*). Significantly, Rec8 exhibited no such bias, suggesting that this chromosome-specific patterning of axis proteins is not dictated by Rec8 (see later sections). The enrichment of Red1 on small chromosomes mirrored a previously noted bias of average meiotic DSB levels, which are also highest on the three shortest chromosomes (*Blitzblau et al., 2007*; *Pan et al., 2011a*; *Thacker et al., 2014*). Indeed, the numbers of axis sites on individual chromosomes were highly correlated with the number of Spo11 cleavage events (Spo11-linked oligonucleotide counts; *Figure 1—figure supplement 1D*). Given that axis proteins are required for the majority of meiotic DSB formation (*Mao-Draayer et al., 1996*; *Schwacha and Kleckner, 1997*; *Xu et al., 1997*; *Kugou et al., 2009*), their preferential enrichment on small chromosomes may contribute to the observed chromosome-size bias of meiotic DSB formation.

Consistent with previous findings, DSB hotspots were largely excluded from sites of axis protein binding (*Blat et al., 2002*; *Pan et al., 2011a*; *Panizza et al., 2011*; *Ito et al., 2014*). Most axis sites detected by ChIP-seq were broad, ranging from 1 to 3 kb (*Figure 1—figure supplement 1E*). Spo11 cleavage activity, as determined by sequencing Spo11-associated DSB ends (Spo11 oligos), was strongly depleted in these sites (*Figure 1—figure supplement 1F*). This pattern further corroborates the notion that DSBs occur preferentially within loop regions that are not bound by axis proteins. As is the case with DSB hotspots, motif discovery did not yield strong consensus sequences associated with Red1 binding. A repetitive sequence pattern consisting of varying numbers of a 3-bp GAN unit was found modestly enriched at Red1 peaks (E-value = 6.6e-6), with maximal enrichment at the summits of Red1 peaks (*Figure 1—figure supplement 1G*, P-value = 5.0e-4). However, this motif was not

predictive of Red1 binding and was only present in 16.1% (125/774) of Red1 peaks, indicating that DNA sequence is at best a minor determinant of axis protein binding.

## Axis proteins preferentially bind to 3′ ends of convergent genes

To better understand the binding preference of axis proteins, we examined where Red1 binds relative to genes. We plotted the average Red1 binding signals with respect to all open reading frames (ORFs). We found that Red1 enrichment was significantly biased toward the ends of ORFs, reaching a summit in the 3′ ends (*Figure 1A*). Consistent with the high signal density on short chromosomes, we found an overall elevated signal along genes situated on short chromosomes. To further interpret the enrichment of axis proteins at 3′ ends of genes, we confined the analysis of binding peaks to 500 bp around Red1 summits. We found that 94.6% (732/774) of Red1 peaks were located at the 3′ ends of genes (250 bp on either side of the stop codon; *Figure 1—figure supplement 1H*). This bias contrasts with DSBs, which are predominantly detected at promoter regions (*Blitzblau et al., 2007*; *Pan et al., 2011a*). Moreover, the Red1 signals were strongly enriched within the intergenic regions of convergent gene pairs as compared to tandem gene pairs (*Figure 1B* and *Figure 1—figure supplement 1H*), a pattern previously described for cohesin-associated sites (*Filipski and Mucha, 2002*; *Lengronne et al., 2004*). Closer inspection of convergent gene pairs in two independently generated data sets revealed that the Red1 binding signal typically consisted of two peaks at convergent 3′ ends, separated by a region of signal depletion (*Figure 1B*). These data indicate that axis protein binding between convergent genes is subject to positional constraints.

To examine whether convergent gene arrangement was required to recruit axis proteins, we inserted a *URA3* gene into the intergenic region between *YKL077W* and *YKL075C*, which created a new convergent gene pair and a tandem gene pair (*Figure 1C*). Strikingly, the enrichment at the 3′ end of the new tandem gene disappeared after the insertion, whereas the peak remained at the newly constructed convergent gene pair. To eliminate possible sequence bias, we inserted *URA3* at the same locus but in the opposite direction. Again, Red1 binding was only detected between the convergent gene pair. These results indicate that convergent gene orientation is both necessary and sufficient for enrichment of axis proteins at this axis association site and demonstrate that DNA sequence is not a major determinant of axis protein binding.

## Transcription focuses meiotic axis association sites

We further investigated the unexpected double peaks of axis protein association between convergent gene pairs. One important feature of the compact yeast genome is that convergent intergenic regions are usually very short, and convergent transcripts frequently overlap (*David et al., 2006*). Accordingly, our analysis of published meiotic mRNA-seq data (*Brar et al., 2012a*) revealed that 79% of convergent transcript pairs exhibited overlapping 3′ UTRs during the early stages of meiosis (*Figure 2A,B*). After ranking the convergent gene pairs according to the extent of overlap of their 3′ UTRs, the presence and spacing of the two Red1 peaks appeared directly correlated with the amount of overlap. Gene pairs with an extensive overlap showed well-separated peaks, whereas gene pairs with little or no overlap exhibited a single peak at the midpoint of their intergenic regions (*Figure 2B*). Red1 binding was maximally enriched ~150 bp downstream of the 3′ ends of the respective transcripts, a behavior also observed for cohesin subunits (*Figure 2C*). To test for a possible role of nucleosomes in axis protein localization, we also plotted nucleosome occupancy (*Pan et al., 2011a*). No overall correlation between nucleosome depletion and the gap between the axis protein binding signals was detected, although there was a decline of nucleosome density at a fraction of the convergent intergenic regions (*Figure 2B*), indicating that nucleosomes do not position Red1. Together, these data imply that the local transcriptional landscape shapes axis association patterns.

We investigated whether transcription levels of the underlying genes drive the distribution of Red1. We first extracted gene pairs in which both genes were highly expressed or both genes were lowly expressed and tested for differences in Red1 signals. This analysis revealed higher and more confined Red1 occupancy between two highly transcribing convergent genes than between gene pairs with less transcriptional activity (*Figure 3A*), indicating that ongoing transcription helps to focus axis proteins between convergent gene pairs. We also noted that the relative heights of the two Red1 peaks between a pair of convergent transcripts often differed substantially. Therefore, we sorted convergent gene pairs based on the relative transcriptional activity of the two genes. This analysis uncovered

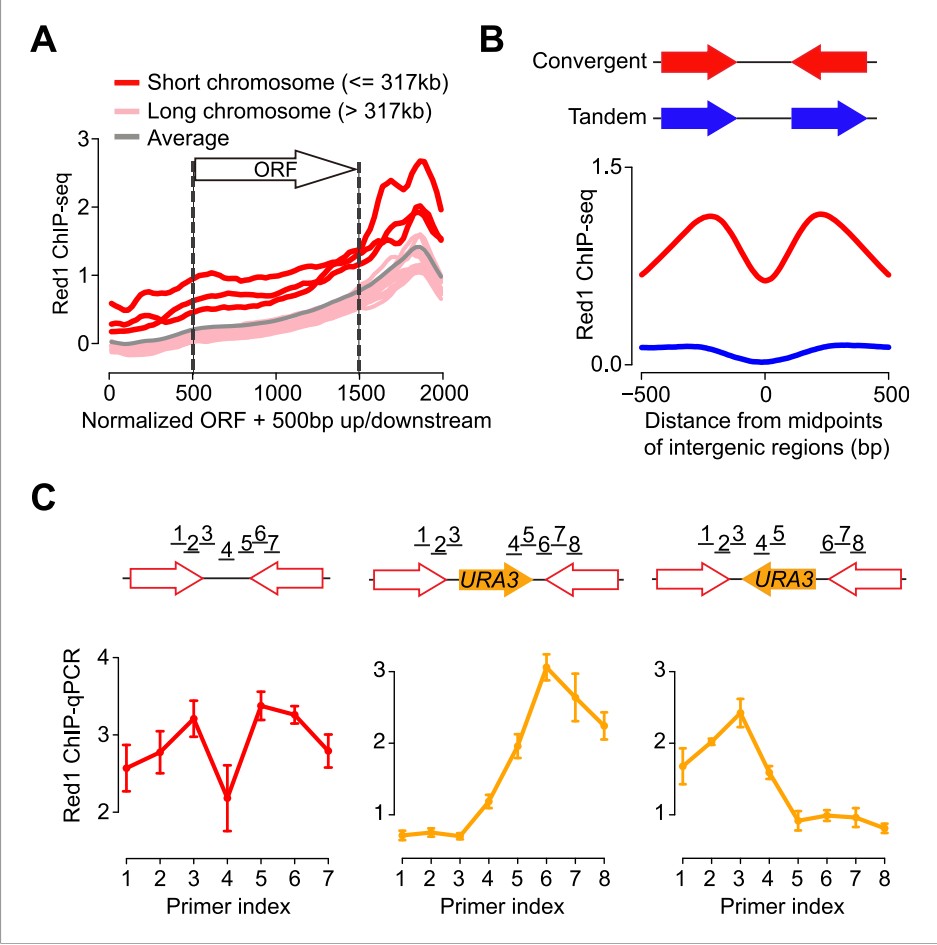

**Figure 1**. Red1 preferentially localizes to the 3′ ends of convergent genes. (**A**) Red1 distribution was plotted as an average across all genes on each of the 16 chromosomes as well as for the whole genome. The coding regions of genes were normalized to lengths of 1 kb (500 bp–1500 bp on x-axis). 500 bp upstream of the start codon and downstream of the stop codon were included in the plot. (**B**) Comparison of Red1 binding between convergent and tandem gene pairs. Average signals of Red1 binding were plotted within a 1 kb region from the midpoints of the intergenic regions. Signals were averaged at each bp among all the convergent (red) and tandem (blue) gene pairs across the genome. (**C**) qPCR analysis of Red1 binding at the *YKL077W/YKL075C* convergent gene pair before and after insertion of a *URA3* transcription unit. Schematic shows the insertion of *URA3* in two orientations between *YKL077W* and *YKL075C*. ChIP signals were normalized against input and an internal control; the control primer was chosen at the promoter region of *YKL077W*. Failure to detect a peak doublet upon *URA3* insertion may be due to primer positions or because the *URA3* transcript does not lead to overlapping transcription (see *Figure 2*).

The following figure supplement is available for figure 1:

**Figure supplement 1**. Genome-wide localization of meiotic axis proteins and cohesin.

a strong reciprocal relationship between transcriptional strength and axis protein enrichment. In gene pairs with differential transcriptional activity, Red1 tends to accumulate downstream of the more highly expressed side. Accordingly, peaks of equal heights were present in gene pairs with similar transcriptional strengths. The reciprocal relationship was apparent both when average Red1 occupancy was calculated for six ranked quantiles and when the sorted gene pairs were displayed as a heatmap (*Figure 3B*). This enrichment pattern suggests that axis proteins are not tightly bound to specific DNA sites but instead are focused between convergent genes driven by some aspect of their transcription.

To further test this possibility, we examined whether changing the level of gene expression is sufficient to alter Red1 binding patterns at axis association sites. We utilized the copper-inducible

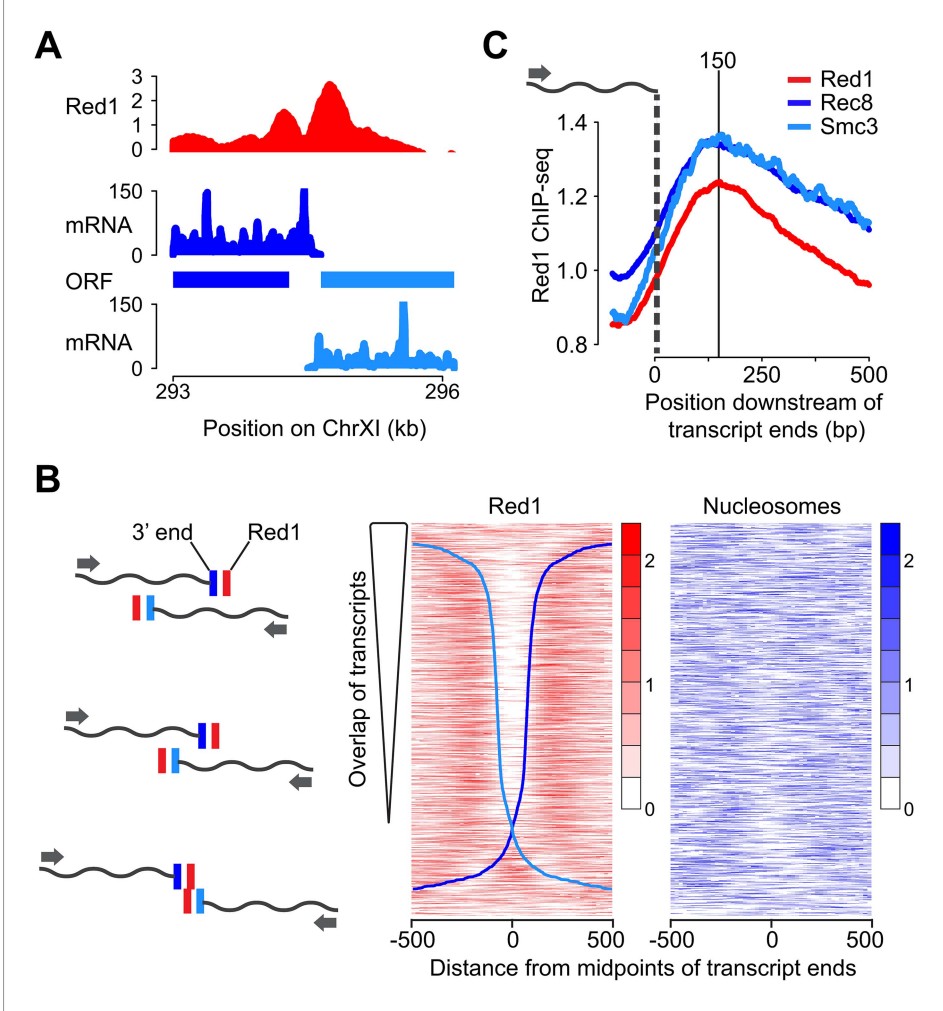

**Figure 2.** Red1 accumulates next to transcript ends of convergent gene pairs. (**A**) Example of overlapping transcripts from a convergent gene pair. ORFs and transcripts of *YKL077W* and *YKL075C* were plotted with respect to their positions on chromosome XI. (**B**) Two-peak pattern of Red1 binding at convergent gene pairs correlates with the amount of transcript overlap. Convergent gene pairs were ranked by the extent of transcript overlap (***Brar et al., 2012a***). Red1 signals (red) were plotted within 1-kb regions centered at the midpoints between of transcript ends. Nucleosome occupancy signals (blue) (***Pan et al., 2011a***) were also plotted as described above. The two blue curves represent the positions of transcript ends of each gene (dark blue = forward direction, light blue = reverse direction; see schematic to the left). (**C**) Enrichment of cohesin subunits and axis proteins was averaged for all convergent transcript ends and plotted as a function of distance from the transcript ends.

*CUP1* promoter (*pCUP1*) to drive the expressions of two genes: *GCY1* and *GAL2*. Both genes exhibit low transcriptional activities during early stages of sporulation and are located in convergent gene pairs, in which the convergently transcribed neighbor is more highly expressed. Two concentrations of copper (5 µM and 20 µM) were applied to induce different levels of expression during early meiosis. In wild-type control cells, the Red1 binding profile was skewed to the sides of the weakly transcribed *GCY1* and *GAL2* genes irrespective of copper concentrations (***Figure 3C,D*** shows *GCY1*; see ***Figure 3—figure supplement 1A,B*** for *GAL2*). A similar profile was also observed for the *pCUP1* strains in the absence of copper, although a shift of Red1 binding away from *pCUP1* was apparent and correlated well with the leaky expression of this promoter (compare ***Figure 3C,D***; also ***Figure 3—figure supplement 1A,B***). Addition of 5 µM $Cu^{2+}$ induced a marked shift in the Red1 binding pattern toward the downstream gene. This shift was further exacerbated upon addition of 20 µM $Cu^{2+}$ (***Figure 3D*** and ***Figure 3—figure supplement 1B***), indicating that the transcription levels

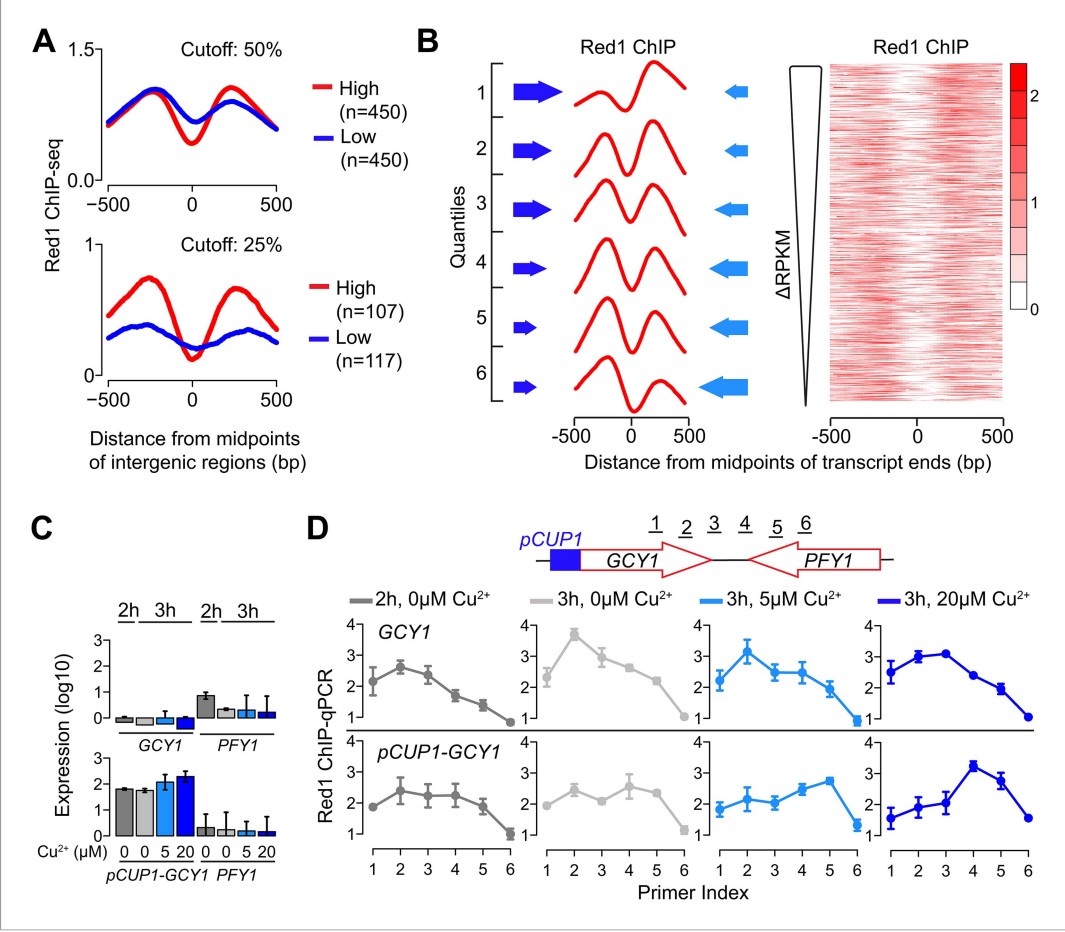

**Figure 3**. Transcriptional activity dictates Red1 binding patterns. (**A**) Convergent gene pairs were identified, in which both genes were strongly transcribed (red) or both genes were weakly transcribed (blue) and average Red1 signal was determined around the midpoints of the intergenic regions. The upper panel represents a cutoff of 50% (both genes were among the top 50% highly transcribed or among bottom 50% lowly transcribed genes); the lower panel represents a cutoff of 25%. (**B**) The Red1 binding signals are biased toward the more weakly transcribed gene at convergent gene pairs. Convergent gene pairs were ranked according to their differences in RPKM values ($\Delta RPKM = RPKM_{forward} - RPKM_{reverse}$), and Red1 binding signals were plotted (right panel). The left panel shows the average Red1 binding signal in six quantiles of ranked $\Delta RPKM$. Arrow sizes schematically represent the relative transcriptional activities of convergent gene pairs. (**C**) Gene expression of the *GCY1/PFY1* convergent gene pair in response to different concentrations of copper was measured by RT-qPCR at the indicated time points. Copper was added to the sporulation medium at 2 hr, and samples were collected at 2 hr and 3 hr. Transcription levels of both convergent genes were examined in a wild-type strain and a strain harboring a *pCUP1* promoter insertion upstream of *GCY1*. Error bars: S.D. of three independent replicates. (**D**) ChIP-qPCR analysis of Red1 binding from the same experiment as in (**C**). Schematic depicts relative position of primer pairs. Error bars: S.D.

The following figure supplement is available for figure 3:

**Figure supplement 1**. Red1 position changes in response to transcription.

of convergent genes underlie the dynamic binding patterns of axis proteins. Such refocusing could either result from sliding motion or repeated recruitment to a chromatin modification or structure at the 3′ ends of transcription bubbles. Previous studies have shown that cohesin is able to change position in response to transcription (*Glynn et al., 2004*; *Lengronne et al., 2004*; *Bausch et al., 2007*). Thus, association with cohesin may explain the transcriptional effects we observe for axis proteins.

## Transcriptional focusing requires cohesin

If cohesin plays a role in the transcription-dependent localization of meiotic axis proteins, these distribution patterns should no longer be apparent when the cohesin ring is disrupted. Our previous analysis had shown that in the absence of meiotic cohesin Rec8, Hop1 still associates with chromosomes but displays large regions of depletion alternating with dense clusters of binding (*Panizza et al., 2011*), a pattern that was also mirrored by Red1 (*Figure 4A*). These large-scale changes in *rec8Δ* mutants were associated with markedly narrower Red1 peaks (*Figure 4—figure supplement 1A*). Importantly, Red1 was no longer biased to the 3′ ends of genes (*Figure 4B*) and focusing of axis proteins at convergent regions was strongly reduced (*Figure 4C*), implying that transcriptional focusing of axis proteins depends on cohesin. By contrast, the small-chromosome bias of axis protein binding persisted in *rec8Δ* mutants (*Figure 4D*), indicating that Red1 is enriched on small chromosomes independently of Rec8.

We investigated the chromosomal landmarks that may direct the formation of axis protein clusters in cohesin mutants. Red1 enrichment over the GAN repeat motif was only modestly increased over background (*Figure 4—figure supplement 1B*), indicating that this motif is not responsible for directing most Red1 binding in cohesin mutants. We also observed no apparent association with

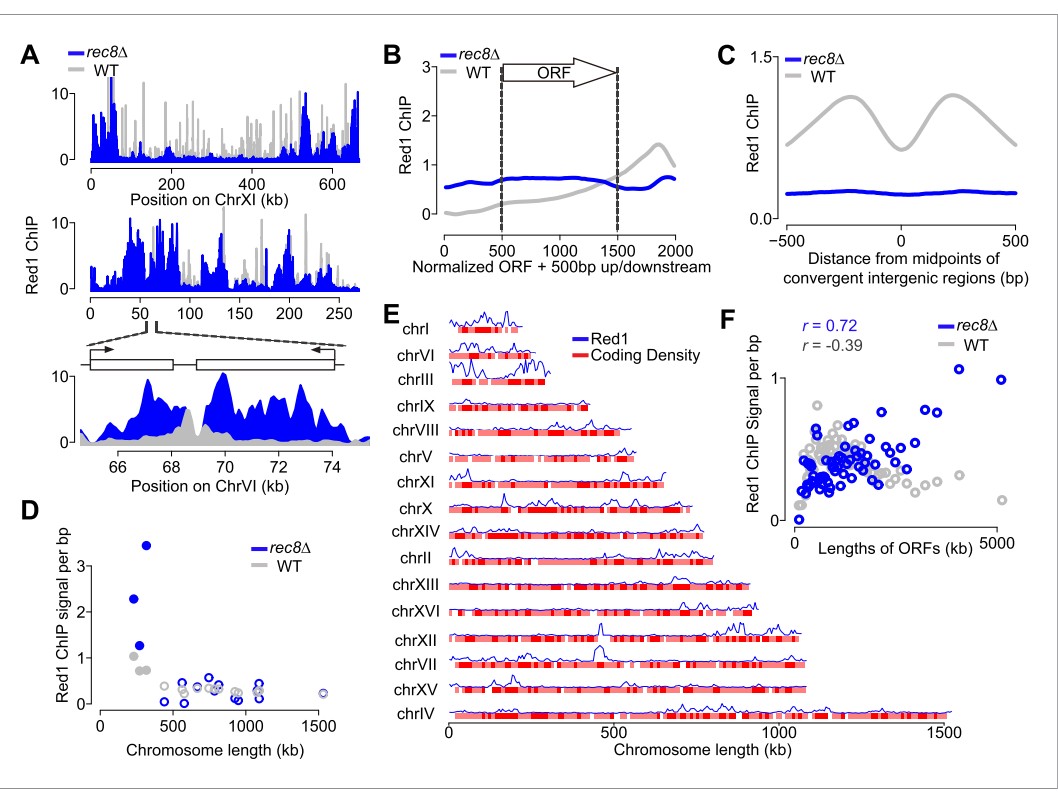

**Figure 4**. Transcriptional focusing depends on cohesin. (**A**) Chromosomal localization of Red1 in *rec8Δ* mutants (blue) and WT (gray) on chromosomes XI and VI. Bottom panel shows zoom-in for a convergent gene pair on chromosome VI. (**B**) Red1 enrichment along genes in *rec8Δ* mutants and WT. (**C**) Average Red1 accumulation in convergent gene regions in *rec8Δ* mutants and WT. (**D**) Average Red1 ChIP signal per bp in *rec8Δ* mutants and WT as a function of chromosome length. (**E**) Alignment of chromosomal Red1 binding in *rec8Δ* mutants (blue) with regions exhibiting high coding density (red). Red1 signals were plotted using a 5-kb smoothing window. Coding density was calculated using 0 (intergenic) or 1 (coding) at each nucleotide position and then smoothed using a 10-kb window and plotted as a heatmap. (**F**) Red1 signal per bp in *rec8Δ* mutants and WT as a function of gene lengths (ORFs). Genes were clustered into 64 groups of 100 genes according to similar gene lengths.

The following figure supplement is available for figure 4:

**Figure supplement 1**. Red1 binding in *rec8Δ* mutants.

nucleotide bias, replication origins, tRNAs, or transposable elements. However, we did note a weak but consistent correlation (Pearson's $r = 0.37$) between axis protein enrichment and the local coding density of the genome. Specifically, regions enriched for long ORFs or tightly clustered short ORFs exhibited increased axis protein association (*Figure 4E* and *Figure 4—figure supplement 1C*). This effect appeared to be cumulative, as longer ORFs displayed higher Red1 binding *per kb* than shorter ORFs (Pearson's $r = 0.70$; *Figure 4F*). Enrichment of Red1 on gene bodies was not apparent in wild-type cells (*Figure 4F*), indicating that cohesin overrides this cohesin-independent axis recruitment activity. Thus, in wild-type cells, Red1 and Hop1 form axis attachment sites by recognizing a feature of cohesin-associated sites, but a cohesin-independent mechanism may contribute additional binding specificity.

## Interdependent binding of Hop1 and Red1 in the absence of cohesin

To further define the mechanisms of axis protein recruitment, we analyzed the mutual dependencies for binding to chromosomes. ChIP-seq analysis of N-terminally tagged V5-Red1 revealed that the majority of Red1 binding sites were unaltered for their positions in the absence of Hop1 (*Figure 5A*), indicating that Hop1 is not essential for recruiting Red1 to axis association sites. We note, however, that Red1 binding appeared to increase at some positions (see next section). Thus, Rec8 determines the positions of Red1 along chromosomes, while Hop1 seems to modulate its amount. Strikingly, Red1 DNA association is largely undetectable in *hop1Δ rec8Δ* double mutants (*Figure 5A*), indicating that Hop1 is required for the cohesin-independent recruitment of Red1. Hop1, on the other hand, is undetectable on chromatin in absence of its partner Red1 even in the presence of Rec8 (*Figure 5B*). Together, these data suggest that Red1 and Hop1 associate with DNA as a complex in the absence of cohesin. By contrast, Hop1 is not required for the recruitment of Red1 to sites of cohesin binding. Finally, with few exceptions, Rec8-HA signals were highly similar in wild-type or *red1Δ* mutants (*Figure 5C*), establishing that cohesin recruits Red1 unilaterally.

## Hop1 modulates accumulation of axis proteins on long chromosomes and near centromeres

Analysis of Red1 binding in *hop1Δ* mutants indicated that peak heights were affected in a non-uniform manner. Specifically, Red1 binding appeared to increase on large chromosomes and close to centromeres. In wild-type cells, we observed higher overall levels of Red1 on the 3 smallest chromosomes (2.5-fold over the rest, *Figure 5D*), correlating well with the higher enrichment of DSB machinery (*Panizza et al., 2011*) and increased DSB levels (*Blitzblau et al., 2007*; *Pan et al., 2011a*) on small chromosomes. The over-representation of Red1 was nearly abolished, falling to 1.25-fold, in the *hop1Δ* mutant (*Figure 5D*). Thus, Hop1 modulates Red1 deposition in a chromosome-dependent manner. Relative scaling of ChIP signals between different experiments by NCIS (Normalization of ChIP-Seq data with untagged control) (*Liang and Keles, 2012*) suggested that the reduced chromosome-size bias may be due to increased Red1 enrichment on the larger chromosomes in the *hop1Δ* mutant. To independently validate the ratio of Red1 occupancy between *hop1Δ* and wild-type strains, we used qChIP at 6 loci with increased and 6 loci with decreased Red1 enrichment. qChIP confirmed the direction of Red1 signal change predicted from ChIP-seq at all 12 genomic positions. *Figure 5E* shows the comparison between the ratios expected from ChIP-seq and those observed by qChIP. These results identify Hop1 as a chromosome-dependent regulator of Red1 deposition and suggest that it down-regulates accumulation of Red1 on the thirteen largest chromosomes.

We also observed increased Red1 binding near centromeres in *hop1Δ* mutants. A doublet of small V5-Red1 peaks was found to flank the yeast centromeres at a stereotypic distance of about 270 bp from their centers (*Figure 5F*). In *rec8Δ* and *rec8Δ hop1Δ* double mutants, these peaks were strongly decreased or missing (*Figure 5—figure supplement 1B*). By contrast, in *hop1Δ* mutants, the centromere proximal peaks of Red1 were strongly enhanced (*Figure 5F*), suggesting that Rec8 recruits Red1 next to the centromere against negative regulation by Hop1, which may in turn possibly prevent unwanted centromere proximal recombination.

The precise positioning of axis proteins at centromeres allowed us to probe the arrangement of cohesin and Red1 within axis association sites in more detail. Whereas the central 200 bp of the budding yeast point centromeres containing the cenH3 (Cse4) hemi-nucleosome (*Henikoff and Furuyama, 2012*; *Henikoff et al., 2014*) were devoid of Rec8, we detected two Rec8 peaks flanking

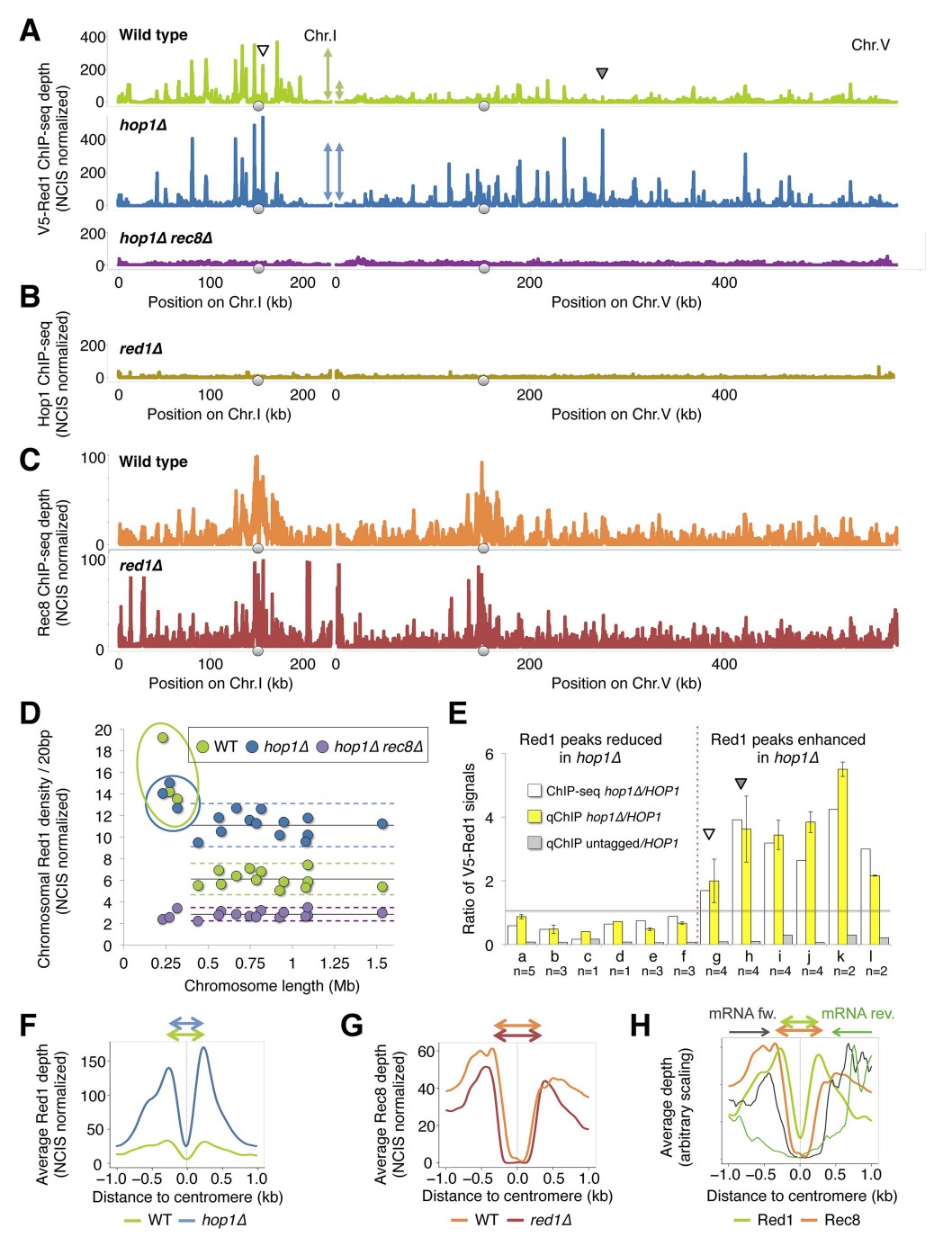

**Figure 5.** Chromosomal localization of Red1 requires Rec8 and Hop1 redundantly. (**A**) Red1 binding profiles in WT (green), *hop1Δ* mutants (blue), and *rec8Δ hop1Δ* mutants (purple) from ChIP-seq experiments (NCIS normalized). A small (Chromosome I) and a medium sized chromosome (Chromosome V) are shown as examples to illustrate the reduced Red1 density on chromosome V in WT but not in *hop1Δ* mutants (indicated schematically by the double-headed arrows). Arrowheads indicate increased Red1 binding confirmed by qPCR in (**E**). (**B**) Genome-wide Hop1 ChIP-seq profiles in *red1Δ* mutants (gold) (NCIS normalized). Chromosomes I and V are shown. (**C**) Genome-wide Rec8 ChIP-seq profiles in WT (orange) and *red1Δ* mutants (red) (NCIS normalized). Chromosomes I and V are shown. (**D**) Red1 density/20 bp (sum of Red1 signal after NCIS normalization divided by chromosomal length) plotted against chromosomal length. The three smallest chromosomes exhibit highly increased binding in WT (green dots), but this difference is largely abolished in *hop1Δ* mutants (blue dots). Red1 density is reduced to noise level and no

*Figure 5. continued on next page*

*Figure 5. Continued*

longer shows biased distribution in *hop1Δ rec8Δ* mutants (purple dots). Horizontal lines indicate the mean of 16 chromosomes (continuous lines), plus minus twofold standard deviation (dashed lines). (**E**) White bars: ratio of V5-red1 ChIP-Seq signal (*hop1Δ/HOP1*) at six cohesin peaks that decrease (a-f: chr3 219k, chr1 171k, chr1 195k, chr6 216k, chr1 95k, chr1 134k) and six cohesin peaks that increase (g-l: chr1 156k, chr5 274k, chr4 435k, chr4 712k, chr7 506k, chr16 576k) upon *HOP1* deletion (data from the experiment shown in (**A**)). Yellow bars: ratio of V5-Red1 signals (*hop1Δ/HOP1*) obtained from ChIP-qPCR primer pairs located at the indicated positions. Gray bars: ratio of V5 signals (untagged/V5-Red1, *HOP1*) obtained at the corresponding positions. The gray line separates reduction (below 1) from increase (above 1). Arrowheads indicate positions on chr1 and chr5 shown in (**A**). DNA from each biological repeat was evaluated both at increasing and decreasing sites to exclude systematic bias in DNA preparation. Each single repeat confirmed the direction of change at each location. n indicates the number of repeats. (**F**) Average densities of V5-Red1 ChIP-seq signals over all 16 yeast centromeres, aligned at their midpoints. The averaged peaks are fourfold higher in *hop1Δ* mutants (blue) than in *HOP1* cells (green). The fold increase at centromeres exceeds the ~twofold increase observed for the entire chromosomes (see **D**). (**G**) Rec8-HA (orange) and Rec8-HA *red1Δ* (red) flank the centromere at a slightly wider distance (around 345 bp) than Red1 (270 bp). (**H**) V5-Red1 (light green) and Rec8-HA (orange) are shown as in (**G**). The average transcription levels from 4 hr after meiotic induction (*Brar et al., 2012a*) around centromeres (forward—black, reverse—dark green) reveal a transcription free pocket, in which Red1 resides. Meiotic cohesin peaks almost coincide with the ends of transcripts. Rec8-HA and V5-Red1 are shown on two different scales.

The following figure supplement is available for figure 5:

**Figure supplement 1**. Analysis of axis protein binding.

the centromeres at roughly 345 bp from the center (*Figure 5G,H*). Rec8 accumulation aligned with the ends of transcripts surrounding the largely transcript-free zone of the centromeres (*Figure 5H*) and was unchanged in *red1Δ* mutants. Red1 peaks are slightly closer to the centromeres (~270 bp), raising the possibility that Red1 is sandwiched between the Cse4 hemi-nucleosome, which may act as a strong barrier, and cohesin, which in turn might be driven by the advancing RNAPII complex.

## Red1, Hop1, and Rec8 physically interact

Given that Rec8 is essential for Red1 recruitment along chromosome arms in *hop1Δ* mutants, we investigated whether the two proteins physically interact. Because interaction between Rec8 and Red1 could preferentially occur in insoluble chromatin, we first used M-track (*Zuzuarregui et al., 2012*) to test their interaction in vivo. M-track uses HKMT, a histone lysine methyltransferase and its acceptor sequence, a histone H3 fragment, each fused to one of the proteins suspected to interact, to create permanent methyl marks on the acceptor even upon transient interaction of the proteins of interest (*Figure 6A*). When HKMT was fused to Rec8, Red1-HA-3xH3 received a robust H3K9me3 signal (*Figure 6B*). The signal existed, as long as Red1 was detectable. We also observed methylation of Hop1-HA-3xH3 by Rec8-HKMT (*Figure 6C*). In agreement with our ChIP-seq data (*Figure 5A*), methyl transfer to Hop1, but not Hop1 protein stability, depended on the presence of Red1 (*Figure 6C*), showing the specificity of the assay. Unexpectedly, Red1-HA-3xH3 depended on Hop1 for its interaction with Rec8 (*Figure 6C*, right panels). This result was at odds with the observation that chromosomal recruitment of untagged Red1 did not require *HOP1* (*Figure 5A*). To investigate this discrepancy, we performed ChIP analysis of Red1 with or without the C-terminal HA-3xH3 tag. Only Red1 tagged at the C-terminus depended on Hop1 for chromosomal binding (*Figure 6D*). Therefore, Hop1 contributes to the interaction between Red1 and Rec8. Moreover, Red1 requires its intact C-terminus when interacting with Rec8 in the absence of Hop1.

The robust signal produced by M-track suggested a stable physical interaction and was in line with previous mass spectrometric detection of Hop1 in a precipitate of Rec8 (*Katis et al., 2010*), although Red1 was not detected in that experiment. To test for Red1-Rec8 interaction more directly, we precipitated V5-Red1 from sonicated and nuclease digested extracts using an anti-V5 antibody and tested the precipitate for the presence of Rec8-HA. Indeed, Rec8-HA co-precipitated and was dependent on the V5-tag of Red1 (*Figure 6E*). Together with the ChIP-seq localization data this results suggests a model in which Rec8 recruits Red1, which further recruits Hop1.

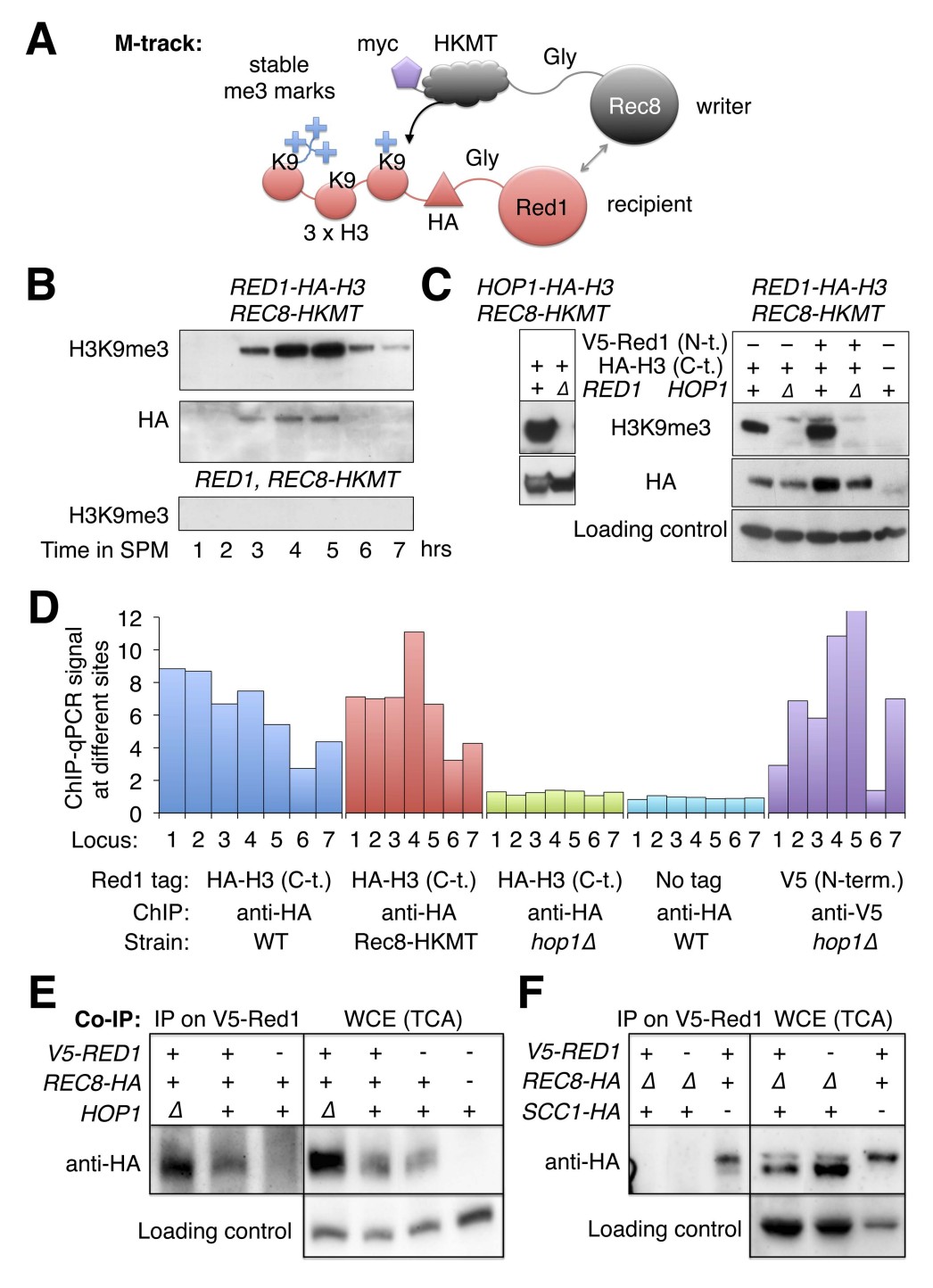

**Figure 6**. Red1 and Hop1 interact with Rec8. (**A**) Illustration of the physical proximity assay M-track. The writer (Rec8) is tagged at on its C-terminus with a human derived histone lysine methyltransferase (HKMT), which will transfer up to three methyl groups to lysine 9 of a small histone H3 fragment presented by the recipient (Red1) depending on the lifespan of the interaction. Using an H3-K9me3 specific antibody, the interaction between writer and recipient can be visualized (*Zuzuarregui et al., 2012*). (**B**) Stable proximity between Red1 and Rec8. Upper panel: detection of interaction between Rec8 and Red1 in a meiotic time course using anti-H3-K9me3 antibody. Middle panel: total Red1-HA-H3 protein using anti-HA antibody. Lower panel: K9me3 antibody signal depends on the H3 tag. (**C**) Left panels: proximity between C-terminally tagged Hop1 and Rec8 depends on Red1. Upper panel: detection of proximity between Rec8 and Hop1 at 4 hr in SPM using anti-H3-K9me3 antibody. Lower panel: total Hop1 protein as
*Figure 6. continued on next page*

*Figure 6. Continued*

determined using anti-HA antibody. Right panels: proximity between C-terminally tagged Red1 and Rec8 depends on Hop1. Upper panel: detection of proximity between Rec8 and Red1 at 4 hr in SPM using anti-H3-K9-me3 antibody. Middle panel: total Red1-HA-H3 protein as determined by Western against the HA tag. Lower panel: loading control. (**D**) When the C-terminus of Red1 is tagged, Hop1 becomes essential for the chromosomal recruitment of Red1. qChIP (using anti-HA antibody) of Red1-HA2-H3 (blue), Red1-HA2-H3 in the presence of Rec8-HKMT (red), Red1-HA2-H3 in *hop1Δ* mutants (green) and untagged Red1 (turquoise) at 7 peak sites (see **Supplementary file 1B**). The values for V5-Red1 in *hop1Δ* mutants (purple) were taken from the profiles shown in **Figure 5A** and put in proportion to Red1 WT (see also qPCR for V5-Red1 in **Figure 5—figure supplement 1A**). (**E**) Left panel: at 4 hr in SPM Red1 was immuno-precipitated with anti-V5 antibody and co-precipitating Rec8 was detected with anti-HA antibody. Lanes from left to right are *V5-RED1 hop1Δ*, *V5-RED1 HOP1*, *RED1 HOP1*. Right panel: Western blot (TCA) of the cultures at the time of the IP. Lanes from left to right are *V5-RED1 hop1Δ*, *V5-RED1 HOP1*, *RED1 HOP1*, *RED1 HOP1*. Rec8 was tagged with HA at the C-terminus except in the rightmost lane, where it was untagged. Swi6 was stained on the stripped blot as a loading control. (**F**) Same type of experiment as in (**E**), but in two strains Scc1-HA was expressed from the *REC8* promoter, while *REC8* was deleted. Lanes from left to right are *V5-RED1 pREC8-SCC1-HA rec8Δ*, *RED1 pREC8-SCC1-HA rec8Δ*, *V5-RED1 SCC1 REC8-HA*. Right panel: Western blot (TCA) of the same cultures at the time of the IP. Swi6 served as loading control.

## Rec8-dependent axis recruitment ensures genome-wide DSB distribution

To investigate the importance of cohesin-mediated axis patterning for meiotic recombination initiation, we analyzed DSB formation in *rec8Δ* mutants. We used microarray-based measurement of DSB-associated single-stranded DNA (ssDNA arrays) to locate DSB hotspots in a *dmc1Δ* mutant background, which prevents DSB repair and allows cumulative DSB measurements (**Blitzblau et al., 2007**; **Buhler et al., 2007**). In the presence of cohesin, DSB hotspots were distributed across most of the genome. By contrast, DSB formation in *rec8Δ* mutants was largely restricted to small chromosomes and the vicinity of Red1 clusters (**Figure 7A,B**). Thus, DSB formation is limited to the genomic neighborhoods that exhibited persistent Red1 and Hop1 binding in *rec8Δ* mutants. These findings are consistent with our previous observation that these regions selectively retain essential components of the meiotic DSB machinery (**Panizza et al., 2011**). Strikingly, despite the loss of transcriptional focusing of axis association sites in the absence of cohesin (**Figure 4C**) and the overall altered patterns of Red1 distribution, DSB formation within Red1 clusters occurred at the same hotspots as in wild-type cells (**Figure 7B**). Thus, hotspot designation does not depend on the precise location of axis binding sites. These data suggest that the general proximity of axis proteins is sufficient to activate DSB hotspots and that the primary role of Rec8 in controlling DSB formation is to broadly distribute axis proteins across meiotic chromosomes.

We asked whether the need for assembling a DSB-competent meiotic chromosome axis could be one reason why cells express a meiosis-specific form of cohesin. For this purpose, we replaced the only meiosis-specific subunit of cohesin, Rec8, with its mitotic counterpart Scc1/Mcd1 by deleting *REC8* and expressing *SCC1* from the *REC8* promoter (*pREC8-SCC1*) (**Toth et al., 2000**). ChIP analysis revealed that Scc1-containing cohesin was targeted to the same chromosomal sites as Rec8-cohesin, indicating that Scc1 can fully substitute for Rec8 in recruiting cohesin to meiotic chromosomes (Pearson's $r = 0.93$, **Figure 7—figure supplement 1**). However, Scc1-cohesin did not efficiently recruit Red1 (**Figure 7C**). As a result, the DSB distribution of *rec8Δ pREC8-SCC1* cells remained similar to *rec8Δ* mutants and unlike wild-type cells (**Figure 7C**). Consistent with this observation, Scc1 failed to detectably precipitate with V5-Red1 when expressed instead of Rec8 from the *REC8*-promoter (**Figure 6F**). We conclude that only meiotic cohesin can efficiently recruit meiotic axis proteins. Surprisingly, *pREC8-SCC1* did restore transcriptional focusing to axis association sites (**Figure 7D**). These data indicate that focusing of axis proteins and axis protein recruitment are separable activities of meiotic cohesin and suggest that Rec8, but not Scc1, provides the necessary Red1 contacts for evenly distributing axis proteins, and thus recombination events, along meiotic chromosomes.

## Discussion

Meiotic chromosomes assume a compact axial architecture while being actively transcribed. Here, we demonstrate that chromatin anchoring of the axial element is adaptable to ongoing transcription, and

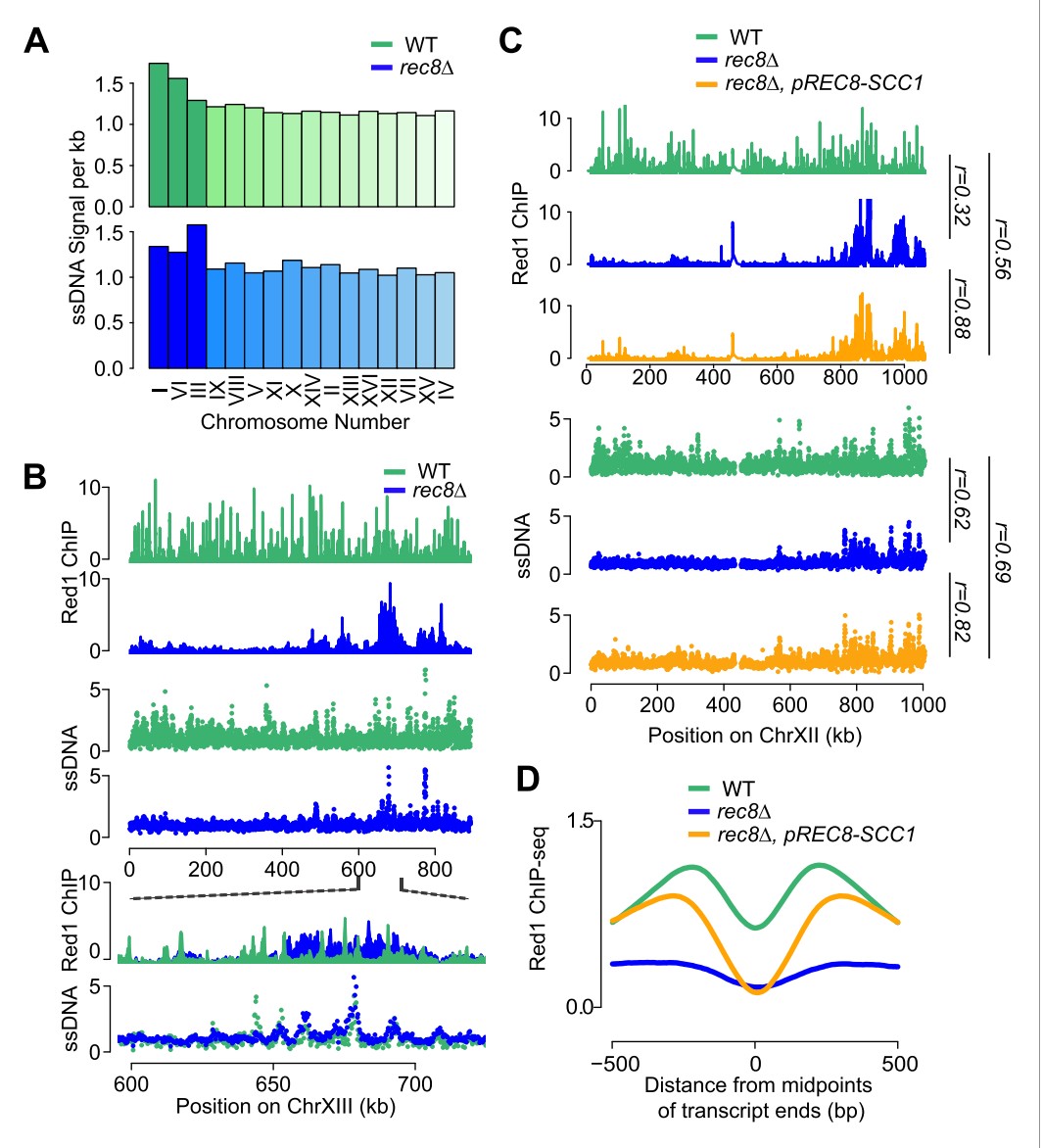

**Figure 7**. Mitotic cohesin rescues transcriptional focusing but not genome-wide defects of *rec8Δ* mutants. (**A**) Relative DSB hotspot intensity of each chromosome in WT (green) and *rec8Δ* mutants (blue) as determined using ssDNA arrays. Chromosomes are sorted by increasing size. (**B**) DSB activity spatially correlates with Red1 enrichment along chromosomes. Distribution of Red1 enrichment (top panels) and DSB hotspots (middle panels) along chromosome XIII in WT (green) and *rec8Δ* mutants (blue). Zoom-in in bottom panels shows that DSB hotspots location is unaltered at finer scales in *rec8Δ* mutants despite severely altered Red1 distribution. (**C**) Ectopic expression of mitotic cohesin does not rescue the defects in Red1 localization or DSB formation of *rec8Δ* mutants. Distribution of Red1 enrichment (top panels) and DSB hotspots (bottom panels) along chromosome XII in WT (green) and *rec8Δ* (blue) and *rec8Δ pREC8-SCC1* (orange). (**D**) Ectopic expression of mitotic cohesin partially rescues transcriptional focusing of Red1 in *rec8Δ* mutants. Average Red1 accumulation in convergent gene regions in WT (green), *rec8Δ* (blue), and *rec8Δ pREC8-SCC1* (orange).

The following figure supplement is available for figure 7:

**Figure supplement 1**. Ectopically expressed Scc1 localizes to the same sites as Rec8 in meiosis.

we identified independent roles for the meiotic cohesin subunit Rec8 and the conserved meiotic axis component Hop1 in guiding axis protein binding throughout the genome and on small chromosomes, respectively. These findings reveal fundamental design principles of higher-order chromosome structure.

## Formation of a loop-axis array

The establishment of a quasi-regular array of chromatin loops represents a considerable mechanistic challenge that could be solved by recruitment motifs or de novo patterning mechanisms. Our findings indicate that the dispersed genomic distribution of cohesin rings at convergent genes (*Glynn et al., 2004*; *Kugou et al., 2009*), and thus ultimately the transcriptional landscape, dictates the preferred attachment sites of the axial element. The enrichment of cohesin between convergent genes is not meiosis-specific (*Glynn et al., 2004*; *Lengronne et al., 2004*), but by replacing the Scc1 subunit of cohesin with a meiotic variant (Rec8), cohesin is converted into a landmark for axial element anchoring. Using oriented *URA3* insertions, we demonstrate that such sequence-independent recruitment ensures that the patterning of axis attachment sites readily adapts to changes in gene arrangements. Moreover, this mechanism may inherently separate axis attachments from DSB hotspots, which occur in promoters and are inhibited by the immediate neighborhood of axis proteins (*Ito et al., 2014*). We note that, similar to yeast, mitotic cohesin is enriched on or near transcribed genes in both flies and mammalian cells (*Wendt et al., 2008*; *Gause et al., 2010*), and cohesin complexes are required for forming the meiotic loop-axis architecture in mammals (*Novak et al., 2008*; *Llano et al., 2012*; *Hopkins et al., 2014*). Thus, the use of the transcriptional landscape as a dynamic reference for meiotic axis assembly may well be conserved through evolution. Perhaps, linking both DSB hotspots and axis association sites to landmarks of active, RNAPII-dependent transcription ensures that meiotic recombination remains confined to euchromatic and thus non-repetitive regions of the genome, reducing the potential for non-allelic homologous recombination and genome instability (*Sasaki et al., 2010*).

Our data show that recruitment of axis proteins involves direct physical interaction with cohesin. The robust positive signal of an in vivo physical proximity assay between Rec8 and Red1 is corroborated by co-precipitation between Red1 and Rec8. Consistent with this observation, Red1 is recruited exclusively to Rec8 binding sites on chromatin by Rec8 in the absence of Hop1. In fact, among cohesin subunits, Rec8 plays a pivotal role, as its mitotic paralogue, Scc1, does not interact notably with Red1. In *Schizosaccharomyces pombe*, meiotic cohesin interacts with the homolog of ScRed1 (Rec10) via Rec11, the meiosis-specific *S. pombe* homolog of ScScc3 (*Sakuno and Watanabe, 2015*). Whether this mechanism of interaction also operates in *S. cerevisiae* is unknown, but as Scc1-cohesin does not recruit or stably interact with Red1 in *S. cerevisiae*, we suggest that Rec8 directly contributes to Red1 binding. These findings indicate that the meiotic cohesin ring forms a stable protein complex with Red1 and Hop1 to flexibly link the axial element with chromatin.

How axis attachment sites congress to form the loop-axis array remains to be determined. Previous data showing that linear elements are lost in the absence of Red1 but remain detectable by electron microscopy (EM) in the absence of Hop1 (*Rockmill and Roeder, 1990*; *Klein et al., 1999*), suggest that Red1, but not Hop1, is required to link axis attachment sites to form linear elements. Red1 has a well-known ability to oligomerize (*Woltering et al., 2000*), which may support the congression of axis sites, although other proteins, including Hop1, may contribute to the wild-type structure. Whether the linear organization emerges from simple nearest neighbor interactions of axis attachment sites along a contiguous piece of DNA remains to be shown.

Another open question is whether all sites of cohesin binding (with the exception of centromeres) function as axis attachment sites on a given chromosome, and thus whether spacing of axis attachment sites can serve as a proxy for estimating loop sizes. Immunofluorescence analyses show Rec8 in continuous tracks, whereas Hop1 and Red1 frequently exhibit more focal staining (e.g., [*Smith and Roeder, 1997*; *Jin et al., 2009*] and data not shown), which may indicate incomplete occupancy of potential axis attachment sites. On the other hand, EM analysis yields estimates of chromatin loop sizes of approximately 20 kb (*Moens and Pearlman, 1988*), which is only moderately larger than the average spacing of cohesin sites of 10.9 kb (*Glynn et al., 2004*), indicating that loops generally do not span more than 2 or 3 potential axis association sites.

## Hop1 quantitatively modulates Red1 accumulation

Hop1 emerges from this work as an intriguing modulator of axis patterning. We show that Hop1 contributes to Red1 recruitment in three ways. First, Hop1 mediates Red1 binding to a number of chromosomal sites independently of Rec8. Second, Hop1 contributes to the recruitment of Red1 by Rec8, as revealed by the loss of Red1–Rec8 interaction in *hop1Δ* mutants when the C-terminus of Red1 is tagged. Third and most unexpectedly, Hop1 is a negative regulator of Red1 accumulation at selected chromosomal regions. It has been noted that DSB formation is dangerous and thus suppressed in the vicinity of centromeres (*Pan et al., 2011a*) and that DSB density is higher on the three smallest chromosomes (*Blitzblau et al., 2007*; *Pan et al., 2011a*), which also enjoy substantially higher meiotic recombination rates (*Kaback et al., 1992*; *Kaback, 1996*). Hop1 strongly suppresses Red1 hyper-accumulation close to centromeres but also on medium and large chromosomes. In the presence of Hop1, Red1 densities on chromosomes I, VI, and III are over twofold higher than on other chromosomes, but without Hop1, Red1 densities for all chromosomes are nearly the same. These findings predict that preferential DSB formation on small chromosomes should be eliminated in a *hop1Δ* mutant. This is indeed the case based on Spo11 oligo mapping (P Schlögelhofer, personal communication). Earlier results have shown that the overrepresentation of DSB machinery on small chromosomes is not dependent on Spo11 (*Panizza et al., 2011*), allowing us to exclude models that involve a DSB feedback mechanism (*Thacker et al., 2014*) to explain the compensatory accumulations. Moreover, the preferential accumulation of Red1 is unaffected, if not exacerbated in *rec8Δ* mutants, consistent with the observation that Rec8 binding itself exhibits no chromosome-size bias. These data exclude a role of cohesin in creating this bias and support the conclusion that Hop1 is the key factor mediating compensatory DSB machine accumulation on small chromosomes.

## The role of axis proteins in activating DSB formation

Our findings support the view that visibly condensed axial structures may not be prerequisite to efficient DSB formation. In particular, yeast *rec8Δ* mutants fail to assemble discernable linear chromosome axes by EM (*Klein et al., 1999*) but remain competent to induce DSBs in chromosomal domains that retain Red1/Hop1 binding. This correlation therefore likely reflects a quasi-local function of axis proteins in hotspot activation that is independent of linear axis formation but this remains to be shown. We previously demonstrated that axis proteins are required for the recruitment of several essential DSB proteins (*Panizza et al., 2011*). Since axis proteins inhibit DSB formation within ~800 bp of their binding sites (*Ito et al., 2014*), the Red1/Hop1-dependent local concentration of DSB factors likely activates hotspots at a distance, involving higher-order chromatin interactions. This model explains why aberrant fine-scale positioning of Red1 in *rec8Δ* mutants still supported DSB formation at wild-type hotspots. The fact that the positions of DSB hotspots remain unchanged in these mutants strongly argues that axis proteins are not responsible for designating certain sequences as hotspots. This conclusion is consistent with growing evidence (summarized in *Borde and de Massy, 2013*) that hotspot designation is primarily driven by epigenetic marks associated with gene promoters.

## Adaptability of the axis-chromatin interface

Significantly, our findings indicate that anchoring of the axial element is highly flexible. We demonstrate that axis protein accumulation peaks ~150 bp downstream of transcript ends and is actively driven by differential promoter activity, implying that axis proteins are not stably bound to DNA. For steric reasons, the observed double peaks likely reflect the average distribution of axis proteins within the assayed cell population, rather than the distribution along a single chromatid. Given that the cohesin ring was previously proposed to slide along chromatin in response to transcription (*Bausch et al., 2007*; *Ocampo-Hafalla and Uhlmann, 2011*), these data indicate that cohesin sliding controls the sites of axis protein attachment. We note that this system allows DNA to be robustly anchored to the protein axis with enough flexibility to allow dynamic processes such as transcription to proceed undisturbed on the DNA (*Figure 8*).

In addition, the adjustable axis-chromatin interface offered by the cohesin ring may also be important during meiotic recombination. Although our data indicate that positional flexibility of axis attachment is dispensable for DSB formation, meiotic chromosome axis components have multiple roles in meiotic recombination, including chromosome pairing, checkpoint signaling and preventing non-productive recombination between sister chromatids (*Zickler and Kleckner, 1999*; *Humphryes*

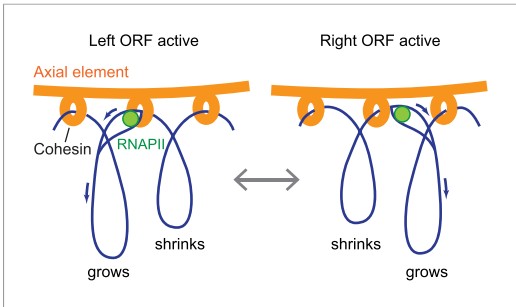

**Figure 8**. Model of the interface between axial elements and DNA. Using the topological protein–DNA interaction of cohesin, a robust linkage between a protein rod (the axial element) and a transcriptionally active chromosome can be established. The protein axis forms on top of cohesin, while the elongating RNAPII, likely unable to pass through the cohesin molecule, may pull the DNA through the complex to avoid a premature transcription block. Blue arrows indicate the direction of DNA flow, going in the opposite direction of transcription. Based on the range of transcriptional overlap, loop movements will in most cases not exceed 500 bp and are not drawn to scale.

and Hochwagen, 2014; *Subramanian and Hochwagen, 2014*). Evidence for recruitment-independent roles of the cohesin ring comes from *rec8Δ* mutants, which are severely defective in maintaining homolog-directed repair bias at a hotspot on chromosome III (*Kim et al., 2010*). This defect occurs despite normal levels of Hop1 and Red1 on this chromosome (*Brar et al., 2009*; *Kugou et al., 2009*; *Panizza et al., 2011*) (this work). Intriguingly, expression of Scc1-cohesin, which we show restores transcriptional focusing for axis proteins, also partially rescues homolog-directed repair bias (*Kim et al., 2010*). Given that targeted DNA repair requires substantial flexibility of DNA ends, these observations raise the possibility that the dynamic anchoring of axis proteins may also be an important prerequisite for controlled meiotic recombination.

## Materials and methods

### Yeast strains and growth conditions

All strains used in this study are of the SK1 background. The genotypes are listed in *Supplementary file 1A*. To induce synchronous meiosis, strains were inoculated at $OD_{600} = 0.3$ in BYTA medium for 16.5 hr at 30°C. Then cultures were washed twice with water and resuspended into SPO medium at $OD_{600} = 1.9$ at 30°C (details are described in *Blitzblau et al., 2012*). For the *pCUP1* experiments, the meiotic culture was split into different flasks 2 hr after meiotic induction. $CuSO_4$ (stock of 50 mM) was then added to induce expression. For experiments shown in *Figures 5, 6* and *Figure 5—figure supplement 1*, strains were diluted 1:2000 from a saturated culture grown for 24 hr in YPD into SPS medium and incubated for 16 hr at 30°C. Then cultures were washed once with SPM medium and resuspended into SPM at $OD_{660} = 1.1–1.2$ at 30°C, shaking 200 rpm/min.

### Chromatin immunoprecipitation

ChIP was performed on samples collected at the 3 hr time point, unless specified otherwise. 25 ml of the meiotic culture was harvested and fixed for 30 min in 1% formaldehyde. The formaldehyde was quenched by addition of 125 mM glycine. Samples were processed as described in *Blitzblau and Hochwagen (2013)*. For ChIP-qPCR, ChIP samples were diluted 1:20 and input samples 1:2000. 8 μl of each dilution were then used in a 20 μl reaction (see RT-qPCR for details). The ChIP samples were quantified by qPCR using two-step normalization. First, the percentage of ChIP relative to input was calculated, followed by calculation of the fold enrichment against an internal control (see *Supplementary file 1B* for primer sequences). For experiments shown in *Figures 5, 6* and *Figure 5—figure supplement 1*, 50 ml of the meiotic culture was harvested at the 4 hr time point, fixed for 15 min in 1% formaldehyde and stopped with 131 mM glycine. For each of the biological replicates in *Figure 5*, at least one increasing and one decreasing locus was included in the assay set to control against variable efficiency of ChIP between the two compared strains. qPCRs in *Figure 6* and *Figure 5—figure supplement 1* differed from previous ones in that the reaction volumes were 25 μl and qPCR at the non-axis locus *ADP1* was used to standardize between different preparations. In *Figure 5—figure supplement 1*, the second normalization factor (*ADP1*) was plotted alongside, instead of being used for normalization.

### ssDNA arrays

The genome-wide analysis of DSB positions using ssDNA enrichment was conducted as described (*Blitzblau and Hochwagen, 2011*).

## Amplification and Illumina sequencing of ChIP DNA

Library preparation was performed using Illumina TruSeq DNA Sample Prep Kits v1 but adapters were used at 1:20 dilution. Amplified ChIP DNA between 300 and 500 bp was gel purified on 1.8% agarose and sequentially quantified using the Qubit assay HS kit as well as an Agilent 2100 Bioanalyzer. Fragments were sequenced on an Illumina HiSeq 2500 instrument using a 51-bp single-end sequencing protocol. For experiments shown in *Figure 5* and *Figure 5—figure supplement 1B*, library preparation was performed according to Illumina ChIP-seq DNA sample prep protocol with TruSeq adapters diluted 1:10. Enzymes for fragment-end polishing and for adapter ligation were from New England Biolabs (Ipswich, MA). Before and after library amplification, ChIP DNA between 200 and 500 bp was gel purified on 2% agarose and quantified using an Agilent 2100 Bioanalyzer. Fragments were sequenced on an Illumina Genome Analyzer using 36-bp single-end sequencing protocols. The raw data analyzed in this study are available from the NCBI Gene Expression Omnibus (http://www.ncbi.nlm.nih.gov/geo/), accession numbers GSE69232 and GSE70112 (*Sun et al., 2015a*, *2015b*).

## Processing of Illumina sequence data

Sequencing reads were mapped to the S288C (sacCer2) genome using Bowtie unless specified otherwise (reads were mapped to the SK1 SGRP genome for the transcriptional analyses [*Brar et al., 2012a*]). Only perfect matches across all 51 bp were considered during mapping. Multiple alignments were not taken into account, which means each read only mapped to one location in the genome. Reads were also mapped to the SK1 genome with similar results. Reads were extended towards 3′ ends to a final length of 200 bp in MACS (http://liulab.dfci.harvard.edu/MACS/) (*Zhang et al., 2008*). For experiments shown in *Figure 5*, sequencing reads were mapped to the S288C (sacCer3) genome using NextGenMap, version 0.4.11 (*Sedlazeck et al., 2013*) (reads were mapped to SK1 SGRP genome for the transcriptional analyses [*Brar et al., 2012a*]), allowing for maximal 2 gaps or 3 mismatches in 36 bp. In case of reads mapping to N locations, the corresponding fraction 1/N was assigned to each location. Reads were extended towards 3′ ends to a final length of in average 139 nt after determining the local optimal extension size (per 10 kb). For comparison between different experiments, profiles were normalized using NCIS (*Liang and Keles, 2012*), after which the untagged control was subtracted. NCIS uses windows of low and randomly distributed sequence counts within each experiment to define the background and to separate it from the signal. Profiles of tagged and untagged samples, as well as before and after background subtraction are deposited at GEO (accession number GSE70112; *Sun et al., 2015b*).

## Peak calling

To identify axis protein enriched regions (peaks), the observed sequence read tag density along the chromosome was normalized against the measured background (Input DNA). Median coverage of all base positions in the ChIP sample was calculated. The coverage at each position was then divided by the calculated median. The same division was also performed for the input sample, followed by subtraction of the input from the ChIP sample at each position. Peak identification was performed using MACS (http://liulab.dfci.harvard.edu/MACS/) (*Zhang et al., 2008*) using a P-value of 1e-15. Peak summits were defined as the local maximum of read coverage within Red1 peaks.

## Motif analysis

Sequences of axis protein binding sites were extracted from the genome using Python scripts (*Supplementary file 2*). We used 500 bp upstream and downstream of the summits to perform motif discovery. All the peaks were ranked by their enrichment signals. An executable version of MDscan (*Liu et al., 2002*) was used with the following parameters: background model: *S. cerevisiae* intergenic; motif width: 6–15; number of top sequences to look for candidate motifs: 30; number of candidate motifs for scanning the rest sequences: 20; number of motifs to report: 5. The online version of MEME-ChIP (http://meme.nbcr.net/meme/cgi-bin/meme-chip.cgi) (*Machanick and Bailey, 2011*) was also used to perform motif discovery on sequencing data with the following parameters: motif site distribution: zero or one occurrence per sequence; count of motifs: 10.

## Proximity assay (M-track)

M-track detects close proximity between proteins by transferring methyl groups via a methyltransferase (HKMT) fused to the bait to an acceptor moiety (histone H3 peptide) fused to the prey. M-track was performed as described by *Zuzuarregui et al. (2012)*. In brief, all fusions were generated by PCR-mediated C-terminal tagging at the endogenous chromosomal locus under the endogenous promoter (genotypes provided in *Supplementary file 1A*). $2 \times 10^{\wedge}8$ cells were collected at the indicated time points. Protein extracts were prepared after trichloroacetic acid treatment as previously described (*Penkner et al., 2005*) and analyzed by Western blotting. Methylated proteins were detected using anti-H3K9me3 antibody (gift from Egon Ogris, 1:100) and anti-HA antibody (12CA5, 1:500) for methylation independent signal as loading control. A non-specific 12CA5 dependent band was used as an additional loading control. Anti-mouse horseradish peroxidase-conjugated antibody (Pierce, 1:10,000) was used as the secondary antibody.

## Co-immunoprecipitation

Co-immunoprecipitation from meiotic culture was performed as described in *de los Santos and Hollingsworth (1999)*. In brief, $2 \times 10^{\wedge}9$ cells were collected at 4 hr in SPM and ruptured using a multibead shocker (YASUI-KIKAI, Osaka) at 2500 rpm, 15 cycles of 30 s on and 30 s off, at 4°C. Extracts were then sonicated using a Covaris S2 (duty cycle: 15%; intensity: 10.0; cycles/burst: 500; total time: 240 s). After removing the cell debris by centrifugation, Benzonase (20 U/sample, Sigma) as well as anti-SV5-coated magnetic beads (50 µl Pan mouse IgG, Dynabeads) were added, followed by rotating incubation for 3 hr at 4°C. Rabbit anti-HA antibody (Sigma, 1:1000) and anti-rabbit horseradish peroxidase-conjugated antibody (Pierce, 1:10,000) were used for detection.

## RNA extraction and RT-qPCR

10 ml of meiotic culture was harvested at the specified time points and total RNA was extracted using glass beads, phenol chloroform and RNA buffer (300 mM NaCl, 10 mM Tris-HCl pH 7.5, 1 mM EDTA, 0.2% SDS). After ethanol precipitation, 10 µg of RNA sample was digested using Turbo DNase (Invitrogen) at 37°C for 30 min followed by adding inactivation mix. We used 1 µg of RNA to generate cDNA, along with dNTP, RNaseOUT Recombinant Ribonuclease Inhibitor (Invitrogen), oligo-dT primer and RevertAid Reverse Transcriptase (Fermentas). Quantitative PCR was performed using the QuantiFast SYBR Green PCR Kit (Qiagen). In a 20 µl of reaction, we used 10 µl 2× SYBR master mix, 1 µl of 20 µM primers, 8 µl of 1:10 diluted cDNA. Primers are listed in *Supplementary file 1B*. Each PCR reaction was performed in triplicate. Results of each sample were normalized against the expression level of an internal control (*GAL1*).

## Acknowledgements

We would like to thank N Hollingsworth, S Roeder and E Ogris for sharing antibodies, G Brar for sharing data prior to publication, D Hur for additional bioinformatics analysis, S Ercan and A-L Kranz for advice on ChIP-seq analysis, and P Schlögelhofer for sharing unpublished results. We are grateful to G Fink for hosting HGB and A v Haeseler for hosting DC for part of this project and the NYU Department of Biology Sequencing Core and the Vienna Biocenter CSF for technical assistance and help with debarcoding. This work was supported by NIH/NIGMS grant R01 GM088248 and Research Grant 6-FY13-105 from the March of Dimes Foundation to AH, a Charles A. King Postdoctoral Fellowship to HGB, as well as grants F3405, F3410 and W1238-B20 from the Austrian Science foundation to FK.

## Additional information

### Funding

| Funder | Grant reference | Author |
| --- | --- | --- |
| National Institutes of Health (NIH) | R01 GM088248 | Andreas Hochwagen |
| March of Dimes Foundation | Research Grant 6-FY13-105 | Andreas Hochwagen |

| Funder | Grant reference | Author |
| --- | --- | --- |
| Austrian Science Fund (FWF) | F3405 | Franz Klein |
| Austrian Science Fund (FWF) | F3410 | Franz Klein |
| Charles A. King Trust | Postdoctoral Fellowship | Hannah G Blitzblau |
| Austrian Science Fund (FWF) | W1238-B20 | Franz Klein |

The funders had no role in study design, data collection and interpretation, or the decision to submit the work for publication.

## Author contributions

XS, LH, HGB, Conception and design, Acquisition of data, Analysis and interpretation of data, Drafting or revising the article; TEM, Acquisition of data, Analysis and interpretation of data, Drafting or revising the article; DC, Analysis and interpretation of data, Drafting or revising the article; FK, AH, Conception and design, Analysis and interpretation of data, Drafting or revising the article

# Additional files

### Supplementary files

• Supplementary file 1. Strains and primers. (**A**) Strains used in the work. (**B**) Primers used for qPCR and RT-qPCR.

• Supplementary file 2. Python code for sequence extraction for motif analysis.

### Major datasets

The following datasets were generated:

| Author(s) | Year | Dataset title | Dataset ID and/or URL | Database, license, and accessibility information |
| --- | --- | --- | --- | --- |
| Sun X, Huang L, Markowitz TE, Blitzblau HG, Chen D, Klein F, Hochwagen A | 2015 | Transcription Dynamically Patterns the Meiotic Chromosome-Axis Interface | http://www.ncbi.nlm.nih.gov/geo/query/acc.cgi?acc=GSE69232 | Publicly available at the NCBI Gene Expression Omnibus(Accession no: GSE69232). |
| Sun X, Huang L, Markowitz TE, Blitzblau HG, Chen D, Klein F, Hochwagen A | 2015 | Transcription Dynamically Patterns the Meiotic Chromosome-Axis Interface | http://www.ncbi.nlm.nih.gov/geo/query/acc.cgi?acc=GSE70112 | Publicly available at the NCBI Gene Expression Omnibus(Accession no: GSE70112). |

Standard used to collect data: Dataset reporting followed the MIAME and MINSEQE community reporting standards for microarray and ChIP-seq datasets respectively.

The following previously published datasets were used:

| Author(s) | Year | Dataset title | Dataset ID and/or URL | Database, license, and accessibility information |
| --- | --- | --- | --- | --- |
| Brar GA, Yassour M, Friedman N, Regev A, Ingolia NT, Weissman JS | 2012 | High-resolution view of the yeast meiotic program revealed by ribosome profiling | http://www.ncbi.nlm.nih.gov/geo/query/acc.cgi?acc=GSE34082 | Publicly available at the NCBI Gene Expression Omnibus(Accession no: GSE34082). |
| Mohibullah N, Zhu X, Thacker D, Keeney S | 2014 | Spo11-oligo mapping in zip3 mutants | http://www.ncbi.nlm.nih.gov/geo/query/acc.cgi?acc=GSE48299 | Publicly available at the NCBI Gene Expression Omnibus(Accession no: GSE48299). |
| Pan J, Sasaki M, Kniewel R, Murakami H, Blitzblau HG, Tischfeld SE, Zhu X, Neale MJ, Jasin M, Socci ND, Hochwagen A, Keeney S | 2011 | Whole-genome nucleosome mapping in meiotic diploid S. cerevisiae | http://www.ncbi.nlm.nih.gov/geo/query/acc.cgi?acc=GSE26452 | Publicly available at the NCBI Gene Expression Omnibus(Accession no: GSE26542). |

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
