## [Decision Letter]

Thank you for sending your work entitled “Transcription Dynamically Patterns the Meiotic Chromosome-Axis Interface” for consideration at *eLife*. Your article has been favorably evaluated by Randy Schekman (Senior editor), a Reviewing editor, and two reviewers, one of whom, Adèle Marston, has agreed to share her identity.

The Reviewing editor and the other reviewers discussed their comments before we reached this decision, and the Reviewing editor has assembled the following comments to help you prepare a revised submission.

This is a careful study probing the functional distribution of meiotic chromosome axis proteins, building on previous work demonstrating the importance of cohesin. The key findings are: (1) transcription spatially determines the localization of axis proteins along chromosomes due to their association with cohesin. (2) Hop1 modulates (reduces) Red1 association with large chromosomes. (3) Even altered Red1 localization correlates with DSBs nearby.

We have one major concern that we would like you to address:

The conclusion that Hop1 reduces Red1 levels on all but the three smallest chromosomes is exciting but should be strengthened. Even with NCIS normalization it is unclear how quantitative ChIP-Seq is. Therefore, the ChIP-qPCR data in Figure 5—figure supplement 1 is important and this analysis should be expanded to look at more sites. Biological replicates should be shown and untagged controls should be included for all experiments. It would also be helpful to use a complementary technique to ChIP. Is the difference between Red1 localization on the different chromosomes observable cytologically, e.g. on chromosome spreads?

We would also like you to tone down the comment about the use of the M track assay to establish the close proximity between Rec8, Red1 and Hop1. We don't think it can it really be used as evidence of direct contact? For instance, is the interaction detectable using standard co-immunoprecipitation methods (with and without DNase?)? My advice is to not do any more experiments but to tone down the conclusions.

---

## [Author Response]

*The conclusion that Hop1 reduces Red1 levels on all but the three smallest chromosomes is exciting but should be strengthened. Even with NCIS normalization it is unclear how quantitative ChIP-Seq is. Therefore, the ChIP-qPCR data in*
Figure 5—figure supplement 1
*is important and this analysis should be expanded to look at more sites. Biological replicates should be shown and untagged controls should be included for all experiments. It would also be helpful to use a complementary technique to ChIP. Is the difference between Red1 localization on the different chromosomes observable cytologically, e.g. on chromosome spreads*?

As the referees have pointed out correctly, it is one thing to show that Hop1 is a conditional regulator of Red1 deposition (this can be derived from the ChIP-seq data with high certainty), but a more sophisticated problem to compare two ChIP-seq profiles and scale them correctly, relative to each other. Although we had NCIS and one independent control (qChIP) included in the orginial submission, we completely agree that this point needed strengthening. We have added 3 additional biological repeats (n listed and error bars added, Figure 5), each comparing Red1-qChIP between *hop1∆* and *HOP1*, and have included an untagged control for each primer pair. We had of course performed untagged controls for all experiments in the first submission but not always shown them for space reasons (ChIP-seq of the untagged control is required for NCIS calculations and can be found as a separate data set at GEO).

In addition, we have expanded the number of intervals analyzed by qChIP. To further strengthen this analysis, we exploited the fact that ChIP-seq not only identified positions where Red1 was increased in the absence of Hop1 (on long chromosomes), but also a few positions where Red1 was decreased. Using 6 positions with an expected increase, in parallel to 6 positions where a decrease is expected, controls against experimental variation concerning overall yield of an IP. Only if the yield of each IP remained constant between the *HOP1* and *hop1∆*, the ratios would be reproduced. If, for instance ChIP of *HOP1* cells produced overall less DNA, positions supposed to be higher in *HOP1* would fail to meet the expectation. However, in all experiments, the ratios on each position predicted to be below 1 behaved as expected, and those predicted to be above 1 did also. Despite this confirmation, we cautiously interpreted our result in the manuscript as “suggesting” down-regulation by Hop1 on long chromosomes.

Chromosome spreads: This is a very good idea, but unfortunately cytological output is on several levels too different from Chip-seq to allow meaningful comparisons. First, ChIP is an averaging technology, while cytology studies single cells. Second, ChIP specifically monitors protein-DNA binding sites. Cytology does not. Big structures, like axial element fragments may be attached with many or few DNA binding sites, but this will not be distinguishable in cytology. Third, cytology will show any binding, independent of sequence specificity, while completely random attachments will generally not be visible in ChIP-seq profiles.

*We would also like you to tone down the comment about the use of the M track assay to establish the close proximity between Rec8, Red1 and Hop1. We don't think it can it really be used as evidence of direct contact? For instance, is the interaction detectable using standard co-immunoprecipitation methods (with and without DNase?)? My advice is to not do any more experiments but to tone down the conclusions*.

We apologize if we overstated the M-track results. Indeed, we did not see co-immunoprecipitation in earlier experiments between Red1 and Rec8, prompting us to turn to M-track in the first place. However, the strength of the H3K9me3 signal suggested a stable physical interaction that was at odds with not getting a Co-IP signal. Red1-Rec8 interaction could lead to insoluble structures that possibly escaped our IP. Therefore we now used strong sonication followed by Benzonase treatment. This yielded a robust Rec8-HA Co-IP signal, when we precipitated with V5-Red1. We hope the referees agree that the Co-IP confirms physical interaction between cohesin and Red1 in the absence of DNA. This result is important, because it suggests a division of labor, in which Red1 has a more structural role both in organizing cohesin into linear, continuous arrays and in recruiting Hop1, while Hop1 may be more specialized for signal transduction and stimulating Spo11 activity.